# Experimental transmission of *Leishmania* (*Mundinia*) parasites by biting midges (Diptera: Ceratopogonidae)

**Tomas Becvar**[1], **Barbora Vojtkova**[1], **Padet Siriyasatien**[2], **Jan Votypka**[1], **David Modry**[3,4,5], **Petr Jahn**[6], **Paul Bates**[7], **Simon Carpenter**[8], **Petr Volf**[1], **Jovana Sadlova**[1] *

**1** Department of Parasitology, Faculty of Science, Charles University, Prague, Czech Republic, **2** Vector Biology and Vector Borne Disease Research Unit, Department of Parasitology, Faculty of Medicine, Chulalongkorn University, Bangkok, Thailand, **3** Biology Centre, Institute of Parasitology, Czech Academy of Sciences, České Budějovice, Czech Republic, **4** Department of Botany and Zoology, Faculty of Science, Masaryk University, Brno, Czech Republic, **5** Department of Veterinary Sciences/CINeZ, Faculty of Agrobiology, Food and Natural Resources, Czech University of Life Sciences Prague, Prague, Czech Republic, **6** Equine Clinic, University of Veterinary Sciences, Brno, Czech Republic, **7** Division of Biomedical and Life Sciences, Faculty of Health and Medicine, Lancaster University, Lancaster, United Kingdom, **8** Entomology Group, The Pirbright Institute, Pirbright, Surrey, United Kingdom

* sadlovaj@natur.cuni.cz

**Data Availability Statement:** All relevant data are within the manuscript and its Supporting Information files.

## Abstract

*Leishmania* parasites, causative agents of leishmaniasis, are currently divided into four subgenera: *Leishmania*, *Viannia*, *Sauroleishmania* and *Mundinia*. The recently established subgenus *Mundinia* has a wide geographical distribution and contains five species, three of which have the potential to infect and cause disease in humans. While the other *Leishmania* subgenera are transmitted exclusively by phlebotomine sand flies (Diptera: Psychodidae), natural vectors of *Mundinia* remain uncertain. This study investigates the potential of sand flies and biting midges of the genus *Culicoides* (Diptera: Ceratopogonidae) to transmit *Leishmania* parasites of the subgenus *Mundinia*. Sand flies (*Phlebotomus argentipes*, *P. duboscqi* and *Lutzomyia migonei*) and *Culicoides* biting midges (*Culicoides sonorensis*) were exposed to five *Mundinia* species through a chicken skin membrane and dissected at specific time intervals post bloodmeal. Potentially infected insects were also allowed to feed on ear pinnae of anaesthetized BALB/c mice and the presence of *Leishmania* DNA was subsequently confirmed in the mice using polymerase chain reaction analyses. In *C. sonorensis*, all *Mundinia* species tested were able to establish infection at a high rate, successfully colonize the stomodeal valve and produce a higher proportion of metacyclic forms than in sand flies. Subsequently, three parasite species, *L. martiniquensis*, *L. orientalis* and *L.* sp. from Ghana, were transmitted to the host mouse ear by *C. sonorensis* bite. In contrast, transmission experiments entirely failed with *P. argentipes*, although colonisation of the stomodeal valve was observed for *L. orientalis* and *L. martiniquensis* and metacyclic forms of *L. orientalis* were recorded. This laboratory-based transmission of *Mundinia* species highlights that *Culicoides* are potential vectors of members of this ancestral subgenus of *Leishmania* and we suggest further studies in endemic areas to confirm their role in the lifecycles of neglected pathogens.

**Funding:** TB, BV, JS and PV were funded by the Czech Science Foundation (https://gacr.cz/en/; grant number 17-01911S). JV, JS and PV were funded by ERD Funds, project CePaViP (https://ec.europa.eu/regional_policy/en/funding/erdf/, grant No. CZ.02.1.01/0.0/0.0/16_019/0000759). Shipment of Culicoides was funded by Research Infrastructures for the control of vector-borne diseases (Infravec2, https://infravec2.eu/), which has received funding from the European Union's Horizon 2020 research and innovation programme under grant agreement No 731060 by JS, SC. SC was funded by Biotechnology and Biological Sciences Research Council grant, https://bbsrc.ukri.org/funding/, BBS/E/I/00007039. The funders had no role in study design, data collection and analysis, decision to publish, or preparation of the manuscript.

**Competing interests:** The authors have declared that no competing interests exist.

## Author summary

*Leishmania* parasites are causative agents of leishmaniasis, a disease affecting millions of humans worldwide. It is widely accepted that these flagellates are transmitted exclusively by phlebotomine sand flies (Diptera: Phlebotominae). Reservoir hosts and insect vectors for the newly established *Leishmania* subgenus *Mundinia*, however, remain poorly understood. Preliminary evidence from field-based studies discovered biting midges (Diptera: Ceratopogonidae) that were naturally infected by *L.* (*Mundinia*) *macropodum* in Australia. This surprising finding led us to carry out a detailed laboratory study aimed at comparison of the development of all currently known species of the subgenus *Mundinia* in both putative vector families. We found that all five *Mundinia* species developed successfully in *C. sonorensis* and the successful transmission of three *Mundinia* species from infected insects to mice was demonstrated for the first time. This is the first detailed *in vivo* evidence that biting midges can act as competent vectors of *Leishmania* parasites of the subgenus *Mundinia* and has considerable epidemiological implications for control of these neglected pathogens.

## Introduction

Leishmaniases are a group of diseases whose etiological agents are the protozoan parasites *Leishmania* (Kinetoplastida: Trypanosomatidae). *Leishmania* circulate between a wide range of natural reservoir hosts and phlebotomine sand flies (Diptera: Phlebotominae) and most commonly cause zoonotic disease in humans, although occurrence of anthroponotic cycles has also been described [1]. The signs of the human disease range from single self-healing cutaneous lesions, diffuse cutaneous and mucocutaneous forms, to the most severe visceral leishmaniasis, which can be fatal if untreated. Over 20 human infecting *Leishmania* species have been recognized and leishmaniasis is present in more than 80 countries worldwide, with around 1 million new cases of cutaneous leishmaniasis and 50 000 to 90 000 cases of visceral leishmaniasis occurring annually. These data are likely to underestimate the true burden of the disease since cases are most common in countries with a low level of infrastructure and healthcare development, so the majority of the cases remain unreported, in addition to the potential impact of asymptomatic or mild cases of infection [2].

The genus *Leishmania* is currently divided into four subgenera: *Leishmania*, *Viannia*, *Sauroleishmania* and *Mundinia* [3]. Subgenera *Leishmania* and *Viannia* include species most frequently detected in humans such as *L. infantum*, *L. donovani*, *L. braziliensis*, *L. major* and are transmitted by sand flies of genera *Phlebotomus* and *Lutzomyia* (Diptera: Psychodidae). The subgenus *Sauroleishmania* includes species infecting reptiles as the primary host, which are transmitted by sand flies of the genus *Sergentomyia* (Diptera: Psychodidae). Finally, the most recently described subgenus *Mundinia* [3] includes five species previously known as the *L. enriettii* complex. While *L. enriettii* [4] and *L. macropodum* [5–6] have only been detected in wildlife kept in captivity, three others, namely *L. orientalis* [7], *L. martiniquensis* [8–9] and an isolate from Ghana that is still formally undescribed [10], have been detected in humans and have the potential to cause disease.

According to phylogenetic analyses, the *Mundinia* subgenus diverges at the base of a phylogenetic tree of the *Leishmania* genus, which points to the ancestral origin of these parasites [6,10] (Fig 1A). This hypothesis is also supported by a worldwide distribution of the subgenus: *L. enriettii* is present in Brazil, *L. macropodum* in Australia, *L.* sp. strain GH5 in Africa

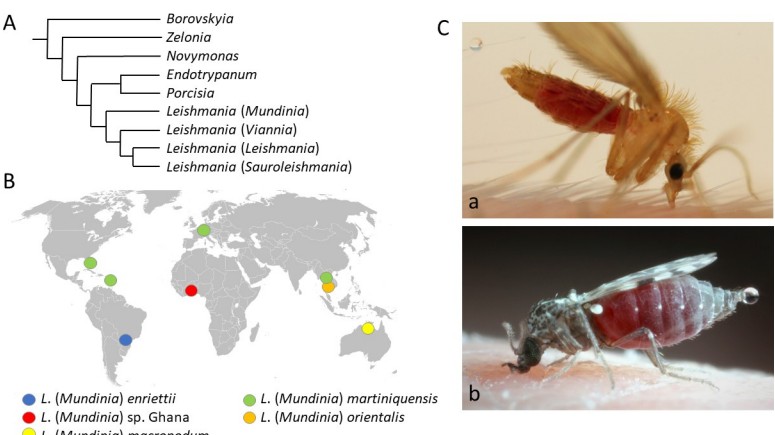

**Fig 1.** (A) Phylogenetic relationships of the four genera of the subfamily Leishmaniinae and subgenera of the genus *Leishmania*, based on [3,6,37]. (B) Geographical distribution of *L.* (*Mundinia*) species, based on [4–5,7–8,10,14–17]. The blank map source - https://commons.wikimedia.org/wiki/Atlasof the_world/. (C) Female sand fly *Phlebotomus duboscqi* feeding on the mouse demonstrating prediuresis which allows rapid concentration of proteins and restoring water and weight balance (a) and *Culicoides* biting midge feeding on a human host while also performing prediuresis (b).

(Ghana) and *L. orientalis* in south-east Asia [11–13] (Fig 1B). *Leishmania martiniquensis* is also a widely distributed species described first from a human case in Martinique island [8], with autochthonous cattle or horse infections reported from Florida [14], Switzerland [15] and Germany [16] and frequently causing human infections in south-east Asia [17–20]. In case of *L. martiniquensis*, however, recent emergence and anthropogenic spread cannot be ruled out, as the infection is known only from humans and domestic animals.

The transmission ecology of the *Mundinia* subgenus is enigmatic, with no certain identity of the reservoir hosts and insect vectors of any species. *Leishmania enriettii* has been isolated from domestic guinea pigs, but subsequent experimental infections of wild guinea pigs as a proposed reservoir species failed [4]. *Leishmania macropodum* was described from four kangaroo species in captivity, but no reports of CL in the wild populations have been published [5,21]. In the remaining species, there is no definitive indication of reservoir or vector, an issue exacerbated by the fact that these may cause asymptomatic infections and hence detection is reliant on either random sampling or experimental studies [22]. Similarly, experimental models enabling research of *Leishmania* pathology are scarce; guinea pigs and golden hamsters were proven to be susceptible to *L. enriettii* [4,23–27] and *L. martiniquensis* was reported to widely disseminate and visceralize in BALB/c mice [28–30]. The only previous study to systematically examine susceptibility to infection for all five *Mundinia* species was carried out in guinea pigs and only *L. enriettii* demonstrated an ability to infect this host [31].

The paradigm that *Leishmania* species pathogenic to humans are transmitted exclusively by phlebotomine sand flies of the genus *Phlebotomus* in the Old World and *Lutzomyia* in the New World has considerable weight in the literature [32–33]. However, observations published in the last decade have raised the possibility that non-sand fly vectors may contribute to the transmission of species within the *Mundinia* subgenus. *Leishmania macropodum* has been detected in biting midges of the genus *Forcipomyia* collected in areas of Australian *Leishmania* transmission [34]. While no *Leishmania*-positive specimens were detected among nearly 2000 sand fly females of four species, DNA was detected in 6% females of three species of *Forcipomyia* and, importantly, heavy late infections were confirmed microscopically [34]. More recently, laboratory experiments have revealed susceptibility of *C. sonorensis* to infection with *L. enriettii* [25] and *L. orientalis* [35].

Based on these studies, biting midges have met three of the four Killick-Kendrick´s criteria necessary to incriminate *Leishmania* vector [36]. However, the most important criterion—the demonstration of transmission by vector bite—was still lacking. Therefore, we compared the development of all 5 currently known *Mundinia* species in biting midge *C. sonorensis* and three sand fly species sharing geographical distribution with respective *Mundinia* species (Fig 1C). Importantly, we demonstrated experimental transmissions of *Leishmania* parasites by biting midges to the host for the first time.

## Results

### *Mundinia* development in sand flies

Sand fly species sharing the geographical area with four *Mundinia* species were used: Brazilian *L. migonei* were infected with *L. enriettii*, Sub-Saharan *P. duboscqi* with *L.* sp. strain GH5 from Ghana and South Asian *P. argentipes* with *L. martiniquensis* and *L. orientalis*. As Australian sand fly species have never been colonized, *L. migonei* permissive to various *Leishmania* species [38–39] was used for *L. macropodum* infections. For each experiment, *Leishmania* species known already to be transmitted by the respective sand fly species was used as the control.

### Development of *L. enriettii* and *L. macropodum* in *Lutzomyia migonei*

Both *Mundinia* species possessed heavy infections (more than 1000 parasites per gut) and high infection rates (more than 80% infected specimens) in sand flies on day 1 post bloodmeal (PBM) only. On day 5 PBM, post defecation by sand fly females, infection rates dropped radically below 9%, and parasites were present in low or moderate numbers (under 100 or 1000 parasites per gut), localized mainly in the abdominal midgut (AMG) or in the beginning of the thoracic midgut (TMG). At eight days PBM, none of the dissected females exposed to *L. enriettii* contained detectable infections and only 2 females infected with *L. macropodum* showed low or moderate infections in AMG and TMG (Fig 2). In contrast, *L. infantum* used as a positive control developed consistently in *L. migonei*, causing 77% infection rate with heavy infections and colonization of the stomodeal valve in 65% of infected females on day 8 PBM (Fig 2).

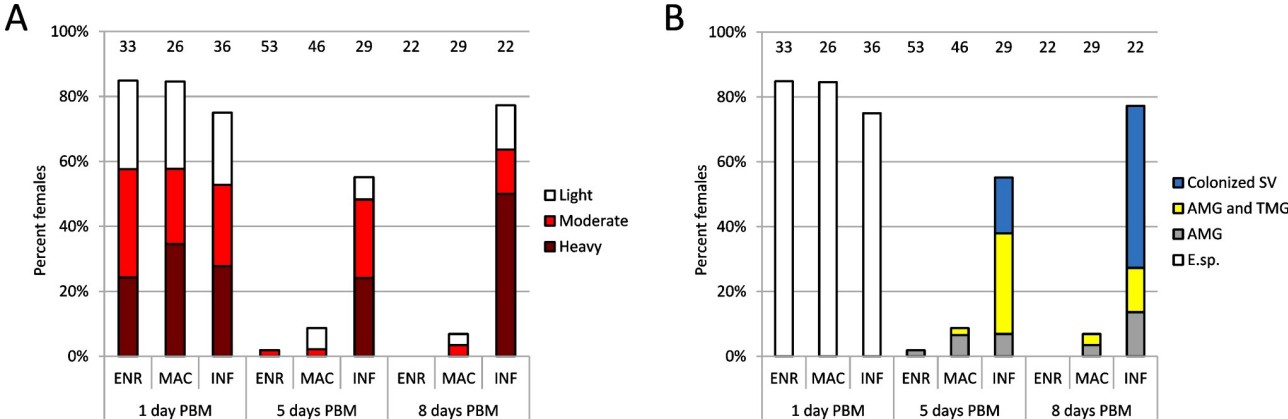

**Fig 2. *Mundinia* development in the sand fly *Lutzomyia migonei*.** Intensity (A) and localization (B) of *L. enriettii* (ENR), *L. macropodum* (MAC) and *L. infantum* (INF) infections. SV, stomodeal valve; AMG, abdominal midgut; TMG, thoracic midgut; E. sp., endoperitrophic space; PBM, post blood meal. Intensity of infection (parasite load) was categorized as light, <100 parasites per gut; moderate, 100–1000 parasites per gut and heavy, >1000 parasites per gut. Numbers of dissected females are displayed above the columns. Statistical differences in intensities of infection among *Leishmania* species were not significant on day 1 PBM (P = 0.860, X$^2$ = 2.571, d.f. = 6) while significant on day 5 PBM (P < 0.0001, X$^2$ = 49.922, d.f. = 6) and day 8 PBM (P < 0.0001, X$^2$ = 44.950, d.f. = 6).

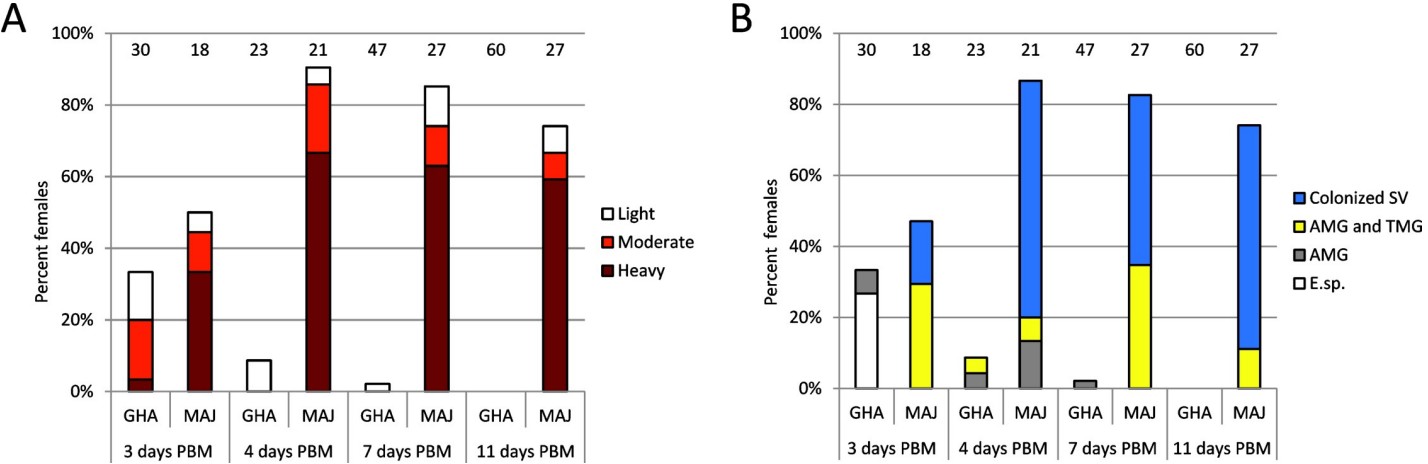

**Fig 3. *Mundinia* development in the sand fly *Phlebotomus duboscqi*.** Intensity (A) and localization (B) of *L.* sp. strain GH5 from Ghana (GHA) and *L. major* (MAJ) infections. SV, stomodeal valve; AMG, abdominal midgut; TMG, thoracic midgut; E. sp., endoperitrophic space; PBM, post blood meal. Intensity of infection (parasite load) was categorized as light, <100 parasites per gut; moderate, 100–1000 parasites per gut and heavy, >1000 parasites per gut. Numbers of dissected females are displayed above the columns. Differences among *Leishmania* species were significant and increased from day 3 PBM (P = 0.039, $X^2$ = 8.352, d.f. = 3) to day 4 PBM (P < 0.0001, $X^2$ = 34.008, d.f. = 3), 7 PBM (P < 0.0001, $X^2$ = 54.884, d.f. = 3) and day 11 PBM (P < 0.0001, $X^2$ = 57.711, d.f. = 3).

## Development of *Leishmania* sp. strain GH5 from Ghana in *Phlebotomus duboscqi*

*Leishmania* sp. from Ghana showed low infection rates (33%) on day 3 PBM, during the bloodmeal digestion. Infection rate was reduced substantially by defecation following the bloodmeal to 9% and then was further reduced to zero over time until day 11 PBM. The few surviving parasites (less than 100 per gut) did not migrate anteriorly, remained localised in the AMG or in the beginning of the TMG (Fig 3). Under the same experimental conditions, control *L. major* developed heavy late-stage infections in 60% of *P. duboscqi* females (Fig 3).

## Development of *L. orientalis* and four *L. martiniquensis* strains in *Phlebotomus argentipes*

During early infections, on day 1 PBM, infection rates and parasite loads of *L. martiniquensis* and *L. orientalis* were comparable with the control *L. donovani*. In later time intervals PBM, *L. donovani* developed heavy infections in almost 80% of *P. argentipes* females, while infection rates of all *Mundinia* strains fell with time, less or more markedly. *Leishmania martiniquensis* MAR1 survived blood digestion and subsequent defecation in 39% of sand fly females and colonised the stomodeal valve as early as on day 4 PBM. Heavy late-stage infections (more than 1000 parasites per gut) were observed in 7% of dissected females 8 days PBM (Fig 4A and 4B). *Leishmania martiniquensis* CU1 survived defecation in 19% of females, but the infection rate further decreased to 15% on day 8 PBM. At this time point most infections were light (less than 100 parasites per gut) and did not reach the cardia region. The experimental infections of *P. argentipes* with *L. martiniquensis* CU2 showed the same trend in reductions in parasite load over time. *Leishmania martiniquensis* Aig1 did not survive defecation of blood remnants in *P. argentipes* as all the females dissected on days 4 and 8 PBM were free of visible infection. On the contrary, *L. orientalis* survived the defecation in relatively high proportion (50%) of *P. argentipes* and heavy late-stage infections with the colonization of the stomodeal valve occurred in 22% of dissected females on day 8 PBM (Fig 4A and 4B).

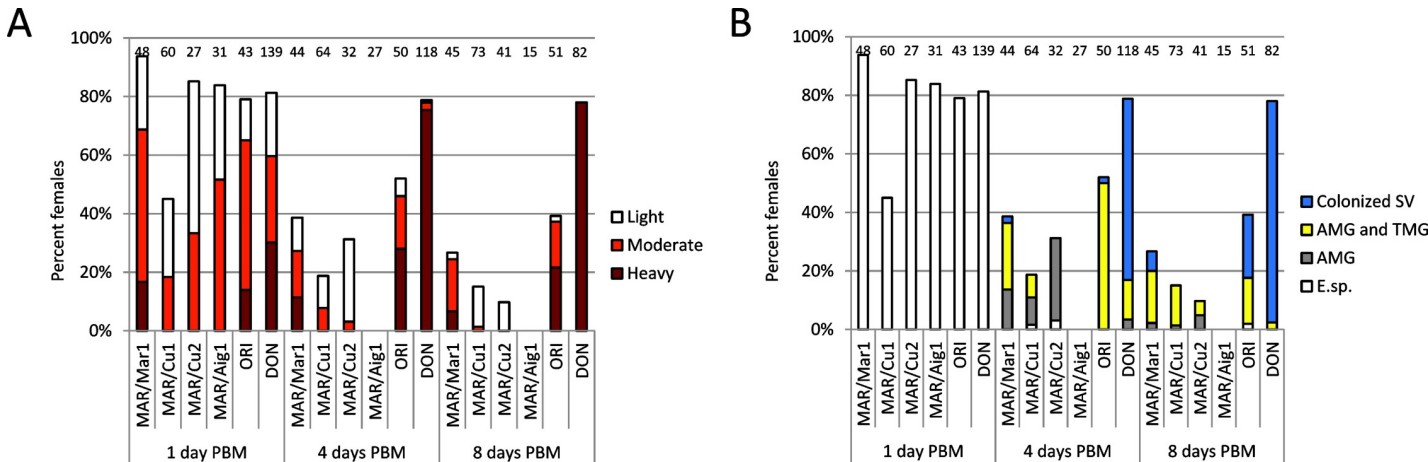

**Fig 4. *Leishmania* development in the sand fly *P. argentipes*.** Intensity (A) and localization (B) of *L. martiniquensis* (MAR; MAR1, Cu1, Cu2, Aig1), *L. orientalis* (ORI) and *L. donovani* (DON) infections assessed by light microscopy. SV, stomodeal valve; AMG, abdominal midgut; TMG, thoracic midgut; E. sp., endoperitrophic space; PBM, post blood meal. Intensity of infection (parasite load) was categorized as light, <100 parasites per gut; moderate, 100–1000 parasites per gut and heavy, >1000 parasites per gut. Numbers of dissected females are displayed above the columns. Differences among *Leishmania* species/strains were significant and increased from day 1 PBM (P < 0.0001, $X^2$ = 97.997, d.f. = 15) to day 4 PBM (P < 0.0001, $X^2$ = 207.642, d.f. = 15) and day 8 PBM (P < 0.0001, $X^2$ = 214.778, d.f. = 15).

## *Mundinia* development in the biting midge *Culicoides sonorensis*

Five *Mundinia* species (8 strains) were tested for development in *C. sonorensis*: *L. enriettii*, *L. macropodum*, *L.* sp. strain GH5 from Ghana, *L. orientalis* and four strains of *L. martiniquensis* (MAR1, CU1, Cu2 and Aig1). On day 1 PBM, the control dissections (3 females per strain) revealed heavy infections and 100% infection rate in all *Leishmania* strains tested. Parasites were enclosed within the peritrophic matrix with the ingested bloodmeal. On day 3 PBM, the digestion of the engorged blood was completed and the bloodmeal remnants passed out with defecation. With the exception of *L. martiniquensis* Aig1, all *Mundinia* species and strains were at least partially successful in surviving defecation: *L. enriettii*, *L. macropodum* and *L. martiniquensis* MAR1 in about 30% of *C. sonorensis* females tested, *L. martiniquensis* Cu2 in more than 50% of females, *L.* sp. strain GH5 from Ghana in 69% and *L. orientalis* and *L. martiniquensis* Cu1 in more than 80% of females (Fig 5A). Parasites were present in the abdominal and thoracic midgut and three species, *L. orientalis*, *L. martiniquensis* Cu2 and *L.* sp. strain GH5 from Ghana, colonized the stomodeal valve in at least some individuals. Hindgut localization of parasites was not observed in any parasite strain (Fig 5B).

By day 6 PBM, the focus of infection had moved anteriorly, the quantity of parasites had increased (Fig 5C) and five (out of eight) strains had colonized the stomodeal valve in at least some *C. sonorensis* examined (Fig 5D). On the other hand, no infections were observed in midges infected with *L. martiniquensis* Aig1 and *L. enriettii*, as parasites did not resist defecation and were either lost or survived in the very small proportion of females. Finally, on day 10 PBM, heavy infections prevailed and the stomodeal valve was colonized in almost all females infected with the 7 strains (Figs 5E and 5F, 6A and 6B). Colonization of the stomodeal valve was usually associated with heavy infection (more than 1000 parasites per gut), however some females had midgut free of parasites, apart from the cardia region packed with haptomonad forms (Fig 6A and 6B). The highest proportion of females *C. sonorensis* with heavy infections including successful colonization of the stomodeal valve were found in *L. orientalis*, *L. martiniquensis* Cu2 and the Ghanaian species, comprising more than 60% of all dissected females (Fig 5E and 5F). Five *Mundinia* species developed significantly heavier late infections in *C. sonorensis* compare to sand fly vector (Table 1).

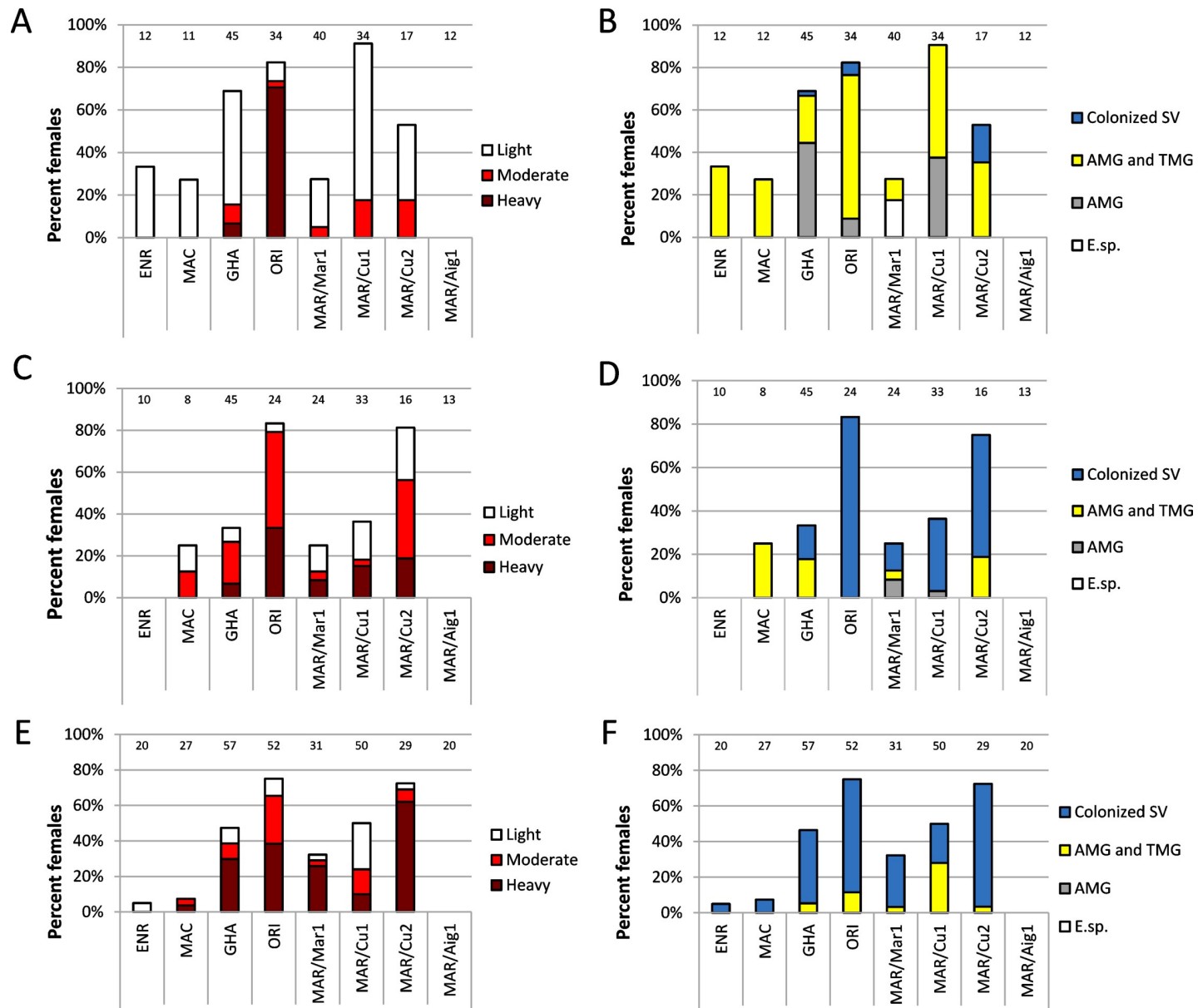

**Fig 5.** *Mundinia* **development in the biting midge** *C. sonorensis.* Intensity (A, C, E) and localization (B, D, F) of *L. enriettii* (ENR), *L. macropodum* (MAC), *L.* sp. strain GH5 from Ghana (GHA), *L. orientalis* (ORI) and *L. martiniquensis* (MAR/MAR1, MAR/Cu1, MAR/Cu2, MAR/Aig1) infections assessed by light microscopy. SV, stomodeal valve; AMG, abdominal midgut; TMG, thoracic midgut; E. sp., endoperitrophic space; PBM, post blood meal. Intensity of infection (parasite load) was categorized as light, <100 parasites per gut; moderate, 100–1000 parasites per gut and heavy, >1000 parasites per gut. Numbers of dissected females are written above the columns. Differences among *Leishmania* species/strains were significant on day 3 PBM (P < 0.0001, $X^2$ = 181.173, d.f. = 21), day 6 PBM (P < 0.0001, $X^2$ = 70.048, d. f. = 21) and day 10 PBM (P < 0.0001, $X^2$ = 117.932, d.f. = 21).

## Metacyclic forms

Presence of infective metacyclic forms was evaluated from Giemsa-stained gut smears in 5 strains showing the high infection rate on day 10 PBM in *C. sonorensis* (*L. martiniquensis* MAR1, Cu1, Cu2, *L. orientalis* and *L.* sp. strain GH5 from Ghana) and in 2 species producing the highest infection rates on day 8 PBM in *P. argentipes* (*L. martiniquensis* MAR1 and *L. orientalis*). Infective stages were detected in *C. sonorensis* infected with *L. martiniquensis*

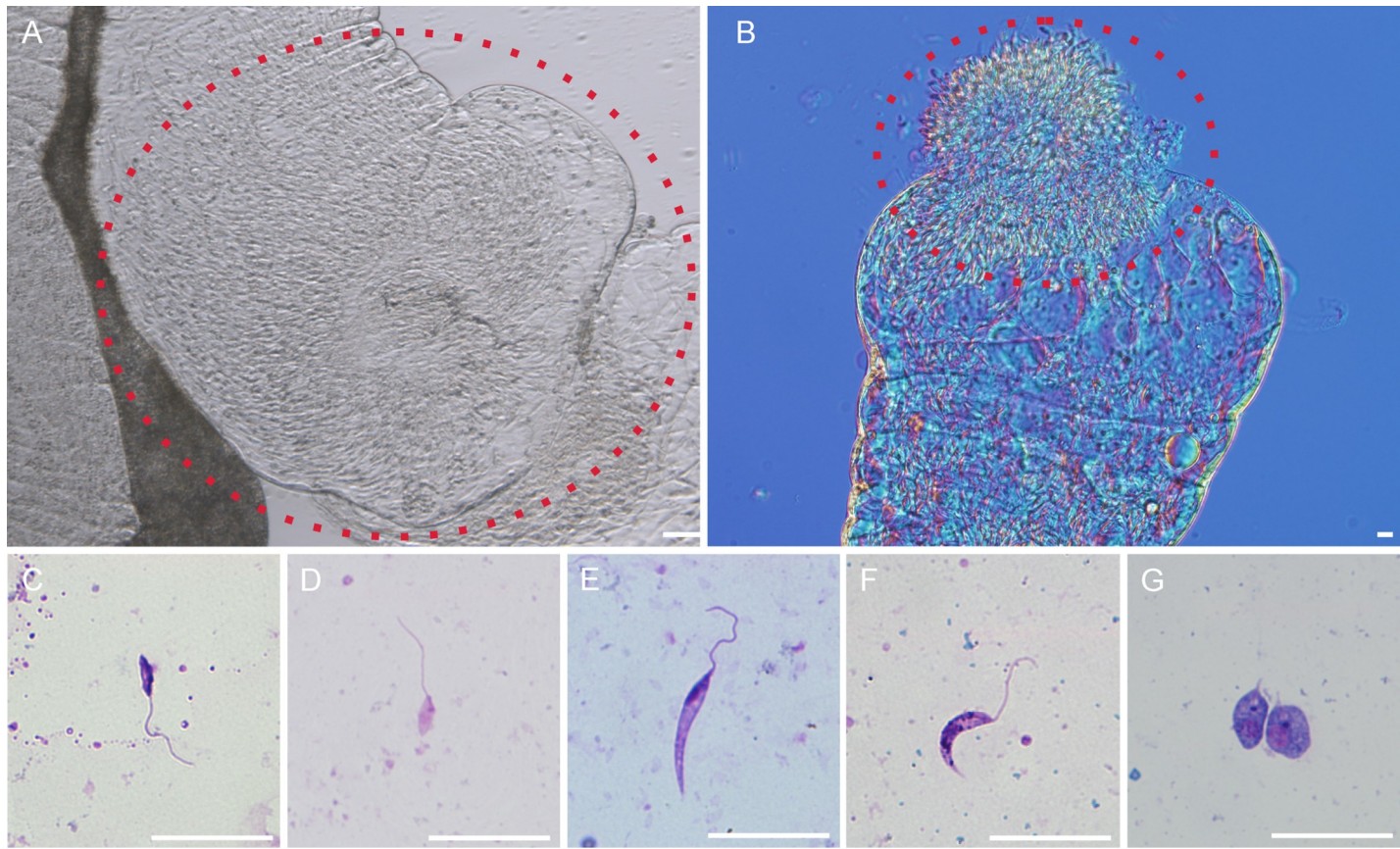

**Fig 6. Localisation and morphology of *Mundinia* in biting midge *C. sonorensis* on day 10 PBM.** A-B, colonization of the stomodeal valve with the part of the thoracic midgut filled with parasites, the region is marked by a red dotted line: (A) mature infection with *L. martiniquensis* Cu1; (B) mature infection with *L. orientalis*. C-G, various morphological forms present in the midgut on day 10 PBM: (C) metacyclic form of *L. martiniquensis* Cu2; (D) metacyclic form of *L. orientalis*; (E) nectomonad form of *L. martiniquensis* MAR1; (F) leptomonad form of *L. martiniquensis* Cu2; (G) haptomonad form of *L. martiniquensis* Cu2. Scale bar = 20μm.

**Table 1. Comparison of intensities of late infections between *C. sonorensis* and sand fly species.**

| *Leishmania* sp. | Vector sp. | N | Statistics |
|---|---|---|---|
| *L. enriettii* | *L. migonei* | 22 | $X^2 = 1.127$, d.f. = 1, P = 0.476 |
| | *C. sonorensis* | 20 | |
| *L. macropodum* | *L. migonei* | 29 | $X^2 = 2.008$, d.f. = 3, P = 0.571 |
| | *C. sonorensis* | 27 | |
| *Leishmania* sp. from Ghana | *P. duboscqi* | 60 | $X^2 = 36.947$, d.f. = 3, P < 0.0001 |
| | *C. sonorensis* | 57 | |
| *Leishmania martiniquensis* Cu1 | *P. argentipes* | 45 | $X^2 = 8.079$, d.f. = 3, P = 0.044 |
| | *C. sonorensis* | 31 | |
| *Leishmania martiniquensis* MAR1 | *P. argentipes* | 73 | $X^2 = 22.099$, d.f. = 3, P < 0.0001 |
| | *C. sonorensis* | 50 | |
| *Leishmania martiniquensis* Cu2 | *P. argentipes* | 41 | $X^2 = 39.595$, d.f. = 3, P < 0.0001 |
| | *C. sonorensis* | 29 | |
| *Leishmania martiniquensis* Aig1 | *P. argentipes* | 15 | Statistics not applicable |
| | *C. sonorensis* | 20 | |
| *Leishmania orientalis* | *P. argentipes* | 51 | $X^2 = 14.271$, d.f. = 3, P = 0.003 |
| | *C. sonorensis* | 52 | |

MAR1 and Cu2, *L. orientalis* and *L.* sp. from Ghana (Table 2), but not with *L. martiniquensis* Cu1 infected midges. The highest representation of metacyclics (5% and 10% respectively) was observed in *C. sonorensis* infected with *L. orientalis* and *L. martiniquensis* Cu2 (Fig 6C and 6D), i.e., the strains which developed heavy late-stage infections in the highest percentage of *Culicoides* females (Fig 5E). The spectrum of other morphological forms produced by *Mundinia* species in *C. sonorensis* is showed in Fig 6E–6G.

In contrast, a lower proportion of metacyclic forms was observed in *Mundinia* infections in sand flies. Two species producing heavy late-stage infections in *P. argentipes* (*L. orientalis* and *L. martiniquensis* MAR1) were analysed. The metacyclics comprised 3% from *L. orientalis* promastigotes (N = 132) and no metacyclic stages were detected in gut smears from females infected with MAR1 (N = 130) 8 days PBM.

## Transmission experiments

Transmission experiments were done with four species of *Leishmania* showing heavy late-stage infections in *C. sonorensis*: *L. orientalis*, *L. martiniquensis*, *L. macropodum* and *L.* sp. strain GH5 from Ghana, and with two species: *L. orientalis* and *L. martiniquensis*, producing heavy late-stage infections in *P. argentipes*. In total, 71 midges and 107 sand flies were allowed to feed on the ear pinnae of anaesthetized BALB/c mice 8–11 days post infective blood meal. Immediately post exposure, mice were sacrificed, their ears stored for PCR analysis, and vector infections were confirmed by microscopical observation. Presence of *Leishmania* infection in engorged vectors was confirmed microscopically in all the experimental groups except in *C. sonorensis* infected with *L. macropodum* (Table 3). Table 3 also illustrates unfed females with mature infections characterized by colonisation of the stomodeal valve, as these females may also contribute at least theoretically to transmission by probing, even without taking a blood meal.

A polymerase chain reaction (PCR) assay with primers flanking a 116 bp segment of the minicircle kinetoplast DNA (kDNA) was used to detect *Leishmania* in mice ear tissues. PCR amplification showed the presence of *Leishmania* minicircle kDNA in mice exposed to *C. sonorensis* infected with two strains of *L. martiniquensis* (Cu1, Cu2), *L. orientalis* and *L.* sp. strain GH5 from Ghana. The samples gathered from transmission experiments performed with *P. argentipes* were negative for both *Mundinia* species in two independent experiments. On the other hand, positive transmission was achieved for control *L. donovani* (S1 Fig and Table 3).

## Discussion

The paradigm that *Leishmania* parasites are transmitted solely by sand flies has been undermined in recent years by the apparent transmission of *L. macropodum* by biting midges of the subgenus *Forcipomyia* (*Lasiohelea*) [34]. Later, successful development of *L. orientalis*, *L. enriettii* and *L. macropodum* was also observed in the biting midge *C. sonorensis* in laboratory

**Table 2. Proportion of metacyclic forms developing in *C. sonorensis* guts 10 days PBM.**

| *Leishmania* strain | Number of measured cells | Percent of metacyclic forms |
|---|---|---|
| *Leishmania martiniquensis* MAR1 | 168 | 0.6 |
| *Leishmania martiniquensis* Cu1 | 214 | 0 |
| *Leishmania martiniquensis* Cu2 | 134 | 10 |
| *Leishmania orientalis* LSCM4 | 175 | 5 |
| *Leishmania* sp. from Ghana GH5 | 180 | 0.6 |

**Table 3. Microscopical examination of *C. sonorensis* and *P. argentipes* females exposed to BALB/c mice for transmission of parasites and result of PCR detection of *Leishmania* minicircle kDNA in mouse ears.**

| Vector species | *Leishmania* strain | Day PBM | No. of females exposed to the mouse | No. of infected engorged females | No. of infected unfed females | No. of unfed females with colonization of the SV | Transmission confirmed by PCR |
|---|---|---|---|---|---|---|---|
| *C. sonorensis* | *L. martiniquensis* MAR1 | 11 | 6 | 1/3 | 2/3 | 2 | No |
| | *L. martiniquensis* Cu1 | 10 | 19 | 2/7 | 5/12 | 5 | Yes |
| | *L. martiniquensis* Cu2 | 11 | 10 | 2/3 | 6/7 | 6 | Yes |
| | *L. orientalis* LSCM4 | 10 | 15 | 7/11 | 4/4 | 4 | Yes |
| | *L.* sp. from Ghana GH5 | 10 | 14 | 4/5 | 7/9 | 7 | Yes |
| | *L. macropodum* | 11 | 7 | 0/3 | 0/4 | 0 | No |
| *P. argentipes* | *L. orientalis* LSCM4 | 8 | 17 | 2/3 | 10/14 | 5 | No |
| | | 8 | 30 | 2/18 | 2/12 | 1 | No |
| | *L. martiniquensis* MAR1 | 8 | 11 | 1/1 | 7/10 | 0 | No |
| | | 8 | 34 | 7/8 | 19/26 | 4 | No |
| | *L. donovani* CUK3 | 8 | 15 | 2/8 | 4/7 | 2 | Yes |

conditions [25,35]. Among sand flies, *Leishmania* of the subgenus *Mundinia* were detected only in Thailand where *L. orientalis* DNA was PCR-detected in *Sergentomyia* (*Neophleboto-mus*) *gemmea* and *S. iyengari* [12,40]. However, without microscopy, it is not possible to distinguish late mature infections from early ones which are non-specific and may be lost with defecation in refractory vectors. Therefore, these molecular findings cannot be considered as a proof of the vector identification [41].

In the current study we have convincingly demonstrated successful infection, propagation and transmission of *Leishmania* species of the subgenus *Mundinia* in *C. sonorensis* under laboratory conditions. In parallel, infection and transmission experiments with epidemiologically relevant sand fly species conducted under the same laboratory conditions and using the same *Leishmania* lines provided only limited evidence of infection and propagation and no evidence of transmission using the *in vivo* mouse model. This study therefore provides the strong underpinning evidence that biting midges may play a role in the transmission of *Leishmania* strains within the subgenus *Mundinia* and highlights the importance of further field-based studies to define this role in areas of pathogen transmission. This is particularly required in the context of the unique geographical distribution of *Leishmania martiniquensis* which is the only *Leishmania* species that occurs in Central or Eastern Europe (the isolate Aig1 used in this study originated from a horse infected either in the Czech Republic or in Ukraine).

*Culicoides sonorensis* is a member of the monophyletic subgenus *Monoculicoides*, which has 24 identified species worldwide and in Europe is represented by five species [42]. A key limitation in research on the *Culicoides* genus as a whole is the lack of availability of epidemiologically relevant colony lines and *C. sonorensis*, which transmits a range of arboviruses in North America, is currently the only major vector species available [43–44]. This limitation is exacerbated by a lack of knowledge regarding the major anthropophilic *Culicoides* species in regions where transmission of *Leishmania* strains occurs, particularly given that these are likely to differ significantly from those adapted for feeding on livestock, both in biology and ecology. The evidence presented in the current paper provides a fundamental reason for studies aimed at

both providing resources for laboratory experimentation and defining transmission ecology of the subgenus *Mundinia*.

Within the studies conducted, *L. enriettii* originating from Brazil was tested in *L. migonei*, a sand fly species widespread in South America [45–46] and known to support development of *L. infantum*, *L. amazonensis* and *L. braziliensis* [38–39]. Our results demonstrated that *L. migonei* possessed limited susceptibility to *L. enriettii*, since none of dissected females developed mature infections. In a majority of females, parasites were defecated with blood remnants and in case parasites survived defecation, they were present in very low numbers in abdominal or thoracic midgut with a very low probability of transmission to the mammalian host. Similar failure of the development was described for *L. enriettii* in *L. longipalpis* [25], the proven vector of *L. infantum* and the most important permissive vector in Latin America [47]. On the other hand, *L. enriettii* colonized the stomodeal valve in 5% of *C. sonorensis*, which corresponds to previously published results [25].

The Ghanaian *Mundinia* species shares the distribution of *P. duboscqi* [48], a proven vector of *L. major* [33]. Our experimental results showed that this member of the *Mundinia* subgenus is not adapted to survive in this sand fly since parasites were lost with defecation of bloodmeal remnants. Nevertheless, this species developed heavy infections and colonized the stomodeal valve in 40% of *C. sonorensis* and were transmitted to mice via feeding on the ear, leading to the potential that this species may also be adapted to biting midge infection and transmission.

*Leishmania macropodum* from Australia did not generate mature infection in *L. migonei* in this study, although it was previously reported to develop in *L. longipalpis* more successfully than *L. enriettii*, showing colonization of the SV in 6% of females [25]. Vector competence of both *Lutzomyia* species tested may be different from those *Sergentomyia* and *Phlebotomus* species that are native to Australia [49], but not available in captivity [50]. In *C. sonorensis*, however, *L. macropodum* colonized the SV in just 7% of females and no infection was detected in females exposed to mice in transmission experiment. However, the transmission may be substantially more effective in biting midges of the subgenus *Lasiohelea*, which were found to be naturally infected in Australia, but which have not been colonised to date [34].

Development of *L. martiniquensis* and *L. orientalis* was assessed in the sand fly species *P. argentipes*, a proven vector of *L. donovani* [51], with a distribution from the Indian peninsula to south-east Asia including Sri-Lanka [52]. We performed experimental infections with *L. orientalis* originating from Thailand and with 4 strains of *L. martiniquensis*—MAR1 originating from the human case in the Martinique island, Cu1 and Cu2 isolated from humans in Thailand and strain Aig1 isolated in the Czech Republic from the four-year-old grey Akchal teke mare horse, imported to the Czech Republic two years ago from Ukraine. Both Thai strains Cu1 and Cu2 survived poorly in *P. argentipes*, failing to develop mature infections, but colonized the SV in 22% and 69% of *C. sonorensis* females, respectively. The strain from the Martinique Island MAR1 generated heavy mature infections in both vector types—colonization of the SV was observed in 7% of *P. argentipes* and 29% of *C. sonorensis* females. Thus, interestingly, the geographically distant isolate of *L. martiniquensis* developed better in *P. argentipes* than sympatric isolates. Importantly, three human *L. martiniquensis* isolates developed late-stage infections in *C. sonorensis* and Thai isolates Cu1 and Cu2 were transmitted to the mouse by *C. sonorensis* bite, while the Aig1 isolate failed to develop.

*Leishmania orientalis* developed heavy late-stage infections with colonization of the SV in 20% of *P. argentipes*, although the same *L. orientalis* strain was reported to be unable to establish infection in *Lu. longipalpis* [35]. In *C. sonorensis*, the SV was colonized even in 63% of females, a much higher rate than previously reported [35]. Metacyclic stages were present in both vector groups, but transmission by bite was demonstrated only for *C. sonorensis*. Thus, the involvement of biting midges in *L. orientalis* transmission is highly likely, although the role

of *Phlebotomus* species was not convincingly excluded. Generally, transmission of *Mundinia* by sand flies must be still considered as various sand fly species present in endemic localities cannot be included into the laboratory study being never colonized, particularly South American and South Asian members of the genus *Sergentomyia*. The only species of the genus *Sergentomyia* where vector competence has been directly tested by experimental infections, *S. schwetzi*, was demonstrated to be refractory to *Leishmania* parasites [53–54]. However, the vector competence of other species of this large genus of sand flies may differ (reviewed by [55]).

One of the basic traits characterizing *Leishmania* subgenera is the mode of their development in the vector. Members of the *Viannia* subgenus undergo peripylarian development, (the hindgut infection is followed by anterior migration of the parasites to the midgut and foregut) while the suprapylarian development in the *Leishmania* subgenus takes place only in the midgut and foregut. Both suprapylarian and peripylarian modes result in transmission by bite contrary to the hypopylarian development in the *Sauroleishmania* subgenus, restricted to the hindgut and resulting in contaminative transmission [56]. In this study, development of all *Mundinia* species tested was purely suprapylarian in both biting midges and sand flies; we did not observe attachment of haptomonad stages in the hindgut, typical for peripylarian development in any sand fly species, nor in *C. sonorensis*. According to the basal position of the subgenus *Mundinia* on the phylogenetic tree of the genus [6], the suprapylarian development may thus represent the ancestral type. However, this finding must be confirmed after identification and colonization of natural vectors of the respective *Mundinia* species.

Within this study, the highest proportion of infective metacyclic forms in mature infections were 10% in *C. sonorensis* infected with *L. martiniquensis* Cu2, 5% of *C. sonorensis* infected with *L. orientalis* and 3% in *P. argentipes* infected with *L. orientalis*. Although these frequencies might look low, similar proportions were reported for mature infections of *L. donovani* in *P. argentipes*; 3–5% of metacyclics were apparently sufficient to transmit the infections to mice [57]. As reported recently, population of metacyclic forms may increase with second and further bloodmeals in natural infections, resulting in greater potential to transmit parasites [58]. Besides metacyclics, other morphological forms are equally important for successful transmission—haptomonads attached to the chitin surface of the stomodeal valve and leptomonads producing promastigote secretory gel create the blocked fly, which forces infected female to regurgitate parasites into the skin (reviewed in [59]).

Taken together, our results strongly suggest potential involvement of biting midges in transmission of *Mundinia* parasites. All five tested species developed better in *C. sonorensis* than in a range of sand fly species, based on survival of vector defecation, the higher rate of the stomodeal valve colonisation and the successful production of metacyclic stages. Most importantly, the transmission to an *in vivo* mouse model was achieved using *C. sonorensis* infected with *L. orientalis*, *L. martiniquensis* and *L.* sp. strain GH5 from Ghana and failed entirely with the sand fly lines used. Adaptation of *Mundinia* to biting midges is most probably specific to this subgenus, as human pathogens *L. donovani*, *L. major* and *L. infantum* of the subgenus *Leishmania* do not establish mature infections in *C. sonorensis* or *C. nubeculosus* [25,60]. Although the results presented here support the significant role of biting midges in *Mundinia* transmission, many aspects of this vector—parasite interaction remain to be resolved.

## Methods

### Ethics statement

BALB/c mice were maintained and handled in the animal facility of Charles University in Prague in accordance with institutional guidelines and Czech legislation (Act No. 246/1992

and 359/2012 coll. on Protection of Animals against Cruelty in present statutes at large), which complies with all relevant European Union and international guidelines for experimental animals. All the experiments were approved by the Committee on the Ethics of Laboratory Experiments of the Charles University in Prague and were performed under permission no. MSMT-7831/2020-3 of the Ministry of Education, Youth and Sports. Investigators are certificated for experimentation with animals by the Ministry of Agriculture of the Czech Republic.

### Sand flies, biting midges and *Mundinia* parasites

Sand fly colonies (*Lutzomyia migonei*, originating from Brazil, *Phlebotomus duboscqi*, originating from Senegal and *Phlebotomus argentipes*, originating from India) were maintained in the Insectary of the Department of Parasitology, Charles University, under the standard conditions (26˚C, humidity in the insectary 60–70%, photoperiod 14 h light/ 10 h dark and fed with 50% sucrose) as described previously [61]. *Culicoides sonorensis* (subgenus *Monoculicoides*) were sent to Charles University from the Pirbright Institute, UK and kept at 25˚C before exposure to feeding. All insects were given free access to 50% sucrose.

*Leishmania enriettii* (MCAV/BR/45/LV90), *L. macropodum* (MMAC/AU/2004/AM-2004), *L.* sp. from Ghana (MHOM/GH/2012/GH5), *L. orientalis* (MHOM/TH/2014/LSCM4), four strains of *L. martiniquensis* (MHOM/MQ/1992/MAR1; MHOM/TH/2011/CU1; MHOM/TH/2019/Cu2 and MEQU/CZ/2019/Aig1), *L. major* (MARV/SN/XX/RV24; LV109) and *L. donovani* s. lat. *(L. infantum/donovani* hybrid (ITOB/TR/2005/TOB2) [62] and *L. infantum* (MHOM/TR/2000/OG-VL)* were used. Parasites were maintained at 28˚C in M199 medium supplemented with 20% foetal calf serum (Gibco, Prague, Czech Republic), 1% BME vitamins (Sigma-Aldrich, Prague, Czech Republic), 2% sterile urine and 250 μg/ml amikacin (Amikin, Bristol-Myers Squibb, Prague, Czech Republic). Before experimental infection, parasites were washed by centrifugation (2400 x g for 5 min), resuspended in saline solution and counted using haemocytometer (Bürker chamber).

### Experimental infections of insects

Female biting midges and sand flies (3–5 days old) were infected by feeding through a chick-skin membrane with promastigotes from log-phase cultures resuspended 1:10 in a heat-inactivated rabbit blood (LabMediaServis) at a final concentration of $1 \times 10^6$ promastigotes/ml. Engorged females were separated and maintained in the same conditions as the colony for subsequent dissections at various time intervals. Intensity and localisation of infection were evaluated under the light microscope; the infections were scored as light (<100 parasites per gut), moderate (100–1000 parasites per gut) or heavy (>1000 parasites per gut) [63]. Differences in intensities of infections were tested by Chi-Square test using the software SPSS version 23. Morphology of parasites from insect guts was evaluated from gut smears fixed with methanol and stained with Giemsa. Promastigotes were examined by light microscopy with an oil immersion objective and photographed using Olympus DP70 camera. Body length and flagellar length of parasites were measured using Image J software. Promastigotes were scored as metacyclic forms when flagellar length ≥ 2 times body length and body length < 14 μm, leptomonad forms when body length < 14 μm and flagellar length > 2 μm and < 2 times body length; nectomonads when body length > 14 μm and haptomonads when flagellum ≤ 2 μm, according to [64].

### Transmission experiments

Experimentally infected insects were maintained for 10 days at 25˚C and then allowed to feed on the naive BALB/c mouse. Animals were anaesthetized with the mixture of ketamin and

xylazine (62 mg/kg and 25 mg/kg). Insect females were placed into small plastic tubes covered with fine mesh and the tubes were held on the ear pinnae of the anaesthetized mouse for one hour. Mice were euthanized immediately post experiment by cervical dislocation under anesthesia. The ear pinnae (the place of biting) were dissected and stored at -20˚C. Insects were dissected immediately post bloodmeal and checked for the presence of *Leishmania* under the light microscope.

### Polymerase chain reaction (PCR) assay

DNA extraction from ear pinnae was performed using the High Pure PCR Template Preparation Kit (Roche) according to the manufacturer's instructions. The total DNA was used as a template for conventional PCR targeting *Leishmania* minicircle kDNA.

Conventional PCR targeting *Leishmania* minicircle kDNA was performed using EmeraldAmp GT PCR Master Mix (TaKaRa) and cycling conditions were as follows: step 1, 94˚C for 3 min; step 2, 94˚C for 20 s; step 3, 63˚C for 20 s; step 4, 72˚C for 5 s; step 5, 72˚C for 5 min; followed by cooling at 12˚C. Steps 2–4 were repeated 40 times. Product length was 116 bp and primers sequences were: Forward- 5´- AGA TTA TGG AGC TGT GCG ACA A- 3´ and Reverse- 5´- TAG TTC GTC TTG GTG CGG TC- 3´ [65]. Samples were analysed using 2% agarose gels.

### Supporting information

**S1 Fig. Amplification of a 116 bp *Leishmania* minicircle kDNA fragment.** 1, Positive control from cultured parasites; 2, Negative control; 3–8, Mouse ear exposed to biting midges (*Culicoides sonorensis*) infected with *L. martiniquensis* Mar1 (3), *L. martiniquensis* Cu1 (4), *L. martiniquensis* Cu2 (5), *L. orientalis* (6), *L.* sp. from Ghana (7), *L. macropodum* (8); 9–13, Mouse ear exposed to *P. argentipes* infected with *L. orientalis* (9, 10), *L. martiniquensis* Mar1 (11, 12) and *L. donovani* (13).
(TIF)

### Acknowledgments

We are grateful to Milena Svobodova for providing *Leishmania donovani/infantum* hybrid and Eric Denison and Jenny Lennon for *Culicoides* production and supply. We also thank to Helena Kulikova, Lenka Krejcirikova and Kristyna Srstkova for the administrative and technical support.

### Author Contributions

**Conceptualization:** Tomas Becvar, Petr Volf, Jovana Sadlova.

**Funding acquisition:** Simon Carpenter, Petr Volf, Jovana Sadlova.

**Investigation:** Tomas Becvar, Barbora Vojtkova, Padet Siriyasatien, Jan Votypka, David Modry, Petr Jahn, Paul Bates, Jovana Sadlova.

**Methodology:** Tomas Becvar, Jovana Sadlova.

**Supervision:** Petr Volf, Jovana Sadlova.

**Writing – original draft:** Tomas Becvar, Jovana Sadlova.

**Writing – review & editing:** David Modry, Paul Bates, Simon Carpenter, Petr Volf.

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
