## [Decision Letter · Decision Letter 0]

10 Mar 2021

Dear Dr. Sadlova,

Thank you very much for submitting your manuscript "Experimental transmission of Leishmania (Mundinia) parasites by biting midges (Diptera: Ceratopogonidae)" for consideration at PLOS Pathogens. As with all papers reviewed by the journal, your manuscript was reviewed by members of the editorial board and by several independent reviewers. In light of the reviews (below this email), several concerns were identified with which I concur.  Both referees were enthusiastic about the work, and thus we would like to invite the resubmission of a significantly-revised version that takes into account the reviewers' comments.   Referee 1 raised important points about missing controls and/or data which will require some attention.

We cannot make any decision about publication until we have seen the revised manuscript and your response to the reviewers' comments. Your revised manuscript is also likely to be sent to reviewers for further evaluation.

Sincerely,

Stephen M. Beverley, Ph.D.

Associate Editor

PLOS Pathogens

David Sacks

Section Editor

PLOS Pathogens

Kasturi Haldar

Editor-in-Chief

PLOS Pathogens

orcid.org/0000-0001-5065-158X

Michael Malim

Editor-in-Chief

PLOS Pathogens

orcid.org/0000-0002-7699-2064

Reviewer's Responses to Questions

**Part I - Summary**

Reviewer #1: This is a well written paper that provides the first definitive experimental evidence that biting midges can transmit Leishmania, sub-genus Mundinia, which includes species such as L. orientalis and L. martiniquensis known to cause human disease. Prior studies had shown that biting midges support the development of Mundinia sp. The current studies extend those findings to include a larger number of Mundinia species and strains, along with parallel infections using colonized species of sand flies that are sympatric with the Mundinia species. The infection parameters described in the sand flies and midges are thorough, and the organization and presentation of the data is excellent, including beautiful photographs of a blood fed sandfly and midge. Mature infections were observed only in the biting midges, with the exception that L. orientalis also produced mature infections in P. argentipes sand flies. The key new finding is that 3 of the tested Mundinia species were successfully transmitted to mice by the bite of the infected midges. By contrast, P. argentipes could not transmit the Mundinia species. An important control involving transmission of L. donovani by P. argentipes was missing.

Reviewer #2: Here, the authors have definitively identified biting midges in the Cullicoides genus as potential natural vectors of parasites in the newly established Leishmania Mundinia subgenus. They convincingly demonstrate successful infection and propagation of five L. Mundinia species in laboratory-reared C. sonorensis, and successful transmission of three of these species to mice via bite (although disease progression in these mice was not examined). While other Leishmania subgenera are transmitted by phlebotomine sand fly bite, the authors convincingly demonstrated that this was not the case for any Mundinia species tested in the laboratory. All experiments were well-designed and included key positive controls. Conclusions are valid and supported by the data presented - with the exception of the “Metacyclic Forms” section in Results, which requires additional data and/or experimental details (see Minor Issues section below).

**Part II – Major Issues: Key Experiments Required for Acceptance**

Reviewer #1: The weakest point of the paper involves the apparent absence of successful transmission of Mundinia sp. by P. argentipes. While this is formally true based on the limited data, the evidence is weak. The possible transmission of L. orientalis by P. argentipes might be expected given the presence of mature infections in 20% of the flies. However, the transmission experience they describe is limited to a single experiment in which only 2 infected, engorged flies were observed, along with 5 unfed flies that showed colonization of the stomodeal valve. More critically, the positive control showing that the colonized P. argentipes flies can transmit L. donovani was not included. By contrast, this control involving P. argentipes infected with L. donovani was included in the comparisons of parasite development in the midgut (fig 2). Since the transmission experiments provide the principle advance of this paper, a more complete study should be shown. Furthermore, the ability of P. argentipes to transmit L. orientalis is itself a biologically relevant question given the sympatric distribution of these sand fly and parasite species in southeast Asia.

Reviewer #2: I have no major problems/issues with this manuscript. However, the “Metacyclic Forms” section in Results requires additional data/information as to how metacyclic forms were identified, as well as how they differ from procyclic forms. The single parasites in Figure 4 panels C and D alone are not sufficient.

**Part III – Minor Issues: Editorial and Data Presentation Modifications**

Reviewer #1: No statistical comparisons of the infection parameters were performed, either between development of the same parasite in midges vs. sand flies, or between different parasite strains in the same vector.

The highest representation of metacyclics in the infected midges was 5 % and 10 %. These are very low frequencies in comparison to the frequencies (>50%) found in sand flies transmitting Leishmania sp. The authors might comment on this point in the discussion.

For the insect colonies used, specify the geographic origin of the wild caught flies used to establish the colony.

Reviewer #2: INTRODUCTION, Lines 66-67: what is meant by “via”? This word infers a logical progression from cutaneous to diffuse cutaneous and mucocutaneous to visceral lesihmaniasis. Is this true?

Line 107: The “G” in guinea pig should be lower-case.

Lines 113 - 115: Say that L. macro was detected in biting midges collected during circulation of disease in red kanagaroos, but Line 100 says there are no reports of CL in wild populations; very confusing as written.

Line 120: “…met three from four Killick-Kendrick’s criteria…”. Maybe replace “from” with “of the”?

RESULTS, Line 143-153: Data showing the failure/poor development of L. Mundinia species in L. migonei is important, and should be included in the body of the paper (not as Supplementary Figure S1).

Line 154-160: Data showing the failure of L. Mundinia species to develop in P. duboscqi is important, and should be included in the body of the paper (not as Supplementary Figure S2). Line 159, might help to ass in the word “control” before “L. major”.

Line 179: Need to define/identify what “LSCM$” and “CUK3” are. I could not find them in Fig. 2.

Line 187-188: Why isn’t Day 1 data presented? How many females were infected on Day 1 post-blood meal? How “heavy” were these infections? Not sufficient to only state all parasites were still enclosed in the blood meal.

Line 197-199: What happened to L. enrietti parasites on Day 6 PBM (Fig. 3 panels C,D)?

Line 203-205: Why was GHA species omitted from this statement? Looks to me that GHA established a heavy infection and successful colonization of the SV, similar to ORI and MAR-Cu2.

Line 215-217: It is very difficult to see the parasites in Figure 4 panel A. An arrow pointing to the parasites, or a dotted line around the parasites, would be very helpful. Easier to see parasites in panel B, but would not hurt to clearly identify the parasites in this panel as well.

Line 219-235: This entire section is lacking crucial information about the criteria used to identify “metacyclic forms”. Furthermore, would be informative to show side-by-side pictures of procyclic and metacyclic forms (rather than the single parasites currently in Figure 4 panels C and D). In Lines 232-233, would be helpful to include the Figure where this data was presented: “…(L. orientalis and L. martiniquensis MAR 1; Fig. 2).”

DISCUSSION, Lines 326 -327: Why do the authors think that geographically distant isolates of L. MAR developed better in P. argentipes than sympatric isolates?

PLOS authors have the option to publish the peer review history of their article (what does this mean?). If published, this will include your full peer review and any attached files.

Reviewer #1: No

Reviewer #2: No
---

## [Editor Report · Decision Letter 1]

18 May 2021

Dear Dr. Sadlova,

We are pleased to inform you that your manuscript 'Experimental transmission of Leishmania (Mundinia) parasites by biting midges (Diptera: Ceratopogonidae)' has been provisionally accepted for publication in PLOS Pathogens.

Best regards,

Stephen Beverley

Associate Editor

PLOS Pathogens

David Sacks

Section Editor

PLOS Pathogens

Kasturi Haldar

Editor-in-Chief

PLOS Pathogens

orcid.org/0000-0001-5065-158X

Michael Malim

Editor-in-Chief

PLOS Pathogens

orcid.org/0000-0002-7699-2064
---

## [Editor Report · Acceptance letter]

2 Jun 2021

Dear Dr. Sadlova,

We are delighted to inform you that your manuscript, "Experimental transmission of Leishmania (Mundinia) parasites by biting midges (Diptera: Ceratopogonidae)," has been formally accepted for publication in PLOS Pathogens.

Best regards,

Kasturi Haldar

Editor-in-Chief

PLOS Pathogens

orcid.org/0000-0001-5065-158X

Michael Malim

Editor-in-Chief

PLOS Pathogens

orcid.org/0000-0002-7699-2064